# Preconception health and care policies, strategies and guidelines in the UK and Ireland: a scoping review protocol

Emma H Cassinelli ![ORCID],[1] Michelle C McKinley ![ORCID],[1] Lisa Kent ![ORCID],[1] Kelly-Ann Eastwood ![ORCID],[1,2] Danielle A J M Schoenaker ![ORCID],[3,4] David Trew,[5] Theano Stoikidou,[5] Laura McGowan ![ORCID] [1]

¹Centre for Public Health, School of Medicine, Dentistry and Biomedical Sciences, Queen's University Belfast, Belfast, UK
²University Hospitals Bristol and Weston NHS Foundation Trust, Bristol, UK
³School of Primary Care, Population Sciences and Medical Education, Faculty of Medicine, University of Southampton, Southampton, UK
⁴NIHR Southampton Biomedical Research Centre, University of Southampton and University Hospital Southampton NHS Foundation Trust, Southampton, UK
⁵Patient and Public Involvement and Engagement, Belfast, UK

**Correspondence to**
Dr Laura McGowan;
laura.mcgowan@qub.ac.uk

## ABSTRACT

**Introduction** Preconception care can significantly improve maternal and infant outcomes, and thus optimise intergenerational health. The aims of this scoping review are to (1) provide an up-to-date summary of preconception health and care strategies, policies, guidelines, frameworks and recommendations across the UK and Ireland and (2) explore preconception health and care services and interventions in Northern Ireland as a case study.

**Methods and analysis** This scoping review of grey literature will be conducted as per the Scoping Review Methods Manual by the Joanna Briggs Institute and the Arksey-O'Malley framework for scoping studies, and reported in line with Preferred Reporting Items for Systematic Reviews and Meta-Analyses extension for Scoping Reviews. Searches were conducted on Google Advanced Search, OpenAire, NICE, ProQuest and relevant public health websites in May 2022. Only results published, reviewed or updated between January 2011 and the time of the searches (May 2022) were considered for inclusion. In addition, searches on interventions and services provided in Northern Ireland will be supplemented by consultations and audits with key stakeholders to validate findings, identify other potentially eligible resources and ensure breadth of coverage. Data will be extracted into Excel and coded using NVivo, and ≥10% of the data will be double-coded. A narrative approach with content analysis highlighting key themes and concepts will be used to report findings.

Throughout the research cycle, members of the wider public will be involved and engaged with to provide feedback.

**Ethics and dissemination** Ethical approval is not required as analyses will be conducted on data available in the public domain. Findings will be shared with relevant stakeholders with the aim to inform future research, practice and decision-making, and disseminated through a peer-reviewed publication, conference presentations and infographics. Dissemination plans will be informed by the 'Healthy Reproductive Years' patient and public involvement and engagement advisory panel.

## STRENGTHS AND LIMITATIONS OF THIS STUDY

⇒ This scoping review will be conducted following a systematic process to transparently locate and synthesise evidence, informed by the Joanna Briggs Institute updated methodological guidance and Arksey and O'Malley's framework, and will be reported in line with the Preferred Reporting Items for Systematic Reviews and Meta-analyses extension for Scoping Reviews.

⇒ This review will incorporate contributions and feedback from those with lived experiences of the preconception period.

⇒ Consultations and audits will be undertaken to validate findings relevant to Northern Ireland, as a case study, identify other potentially eligible resources and understand any under-researched issues concerning the effectiveness of preconception care services or strategies.

⇒ No quality appraisal will be conducted on the included material, as this is a scoping review aiming to map the breadth of information on the chosen topic.

## INTRODUCTION

Preconception health describes the overall health of non-pregnant individuals of childbearing age, which usually defines individuals aged 18–44 years.[1] The optimisation of preconception health can significantly improve maternal and infant outcomes and, therefore, represents a window of opportunity to improve the health of future generations.[2] Preconception care is the term used to define biomedical, behavioural and social health interventions, services, support and advice provided prior to conception aimed at optimising pregnancy planning and fitness for pregnancy.[2 3]

The health status of individuals in the preconception stage can be negatively impacted by risk factors that are often modifiable and/or reversible, and people often have more than one risk factor concurrently.[4] Preconception health risk factors include, for example, parental obesity, long-term physical and mental health conditions, alcohol consumption, smoking, physical inactivity,

inadequate dietary habits, low folic acid intake, poor social support, emotional ill health and low immunisation levels.[2 5 6] While some risk factors are modifiable, such as folic acid intake, others may require further evidence-based support and long-term management to achieve and sustain change, such as weight management in people living with obesity. Risk factors such as financial instability are not easily modifiable and, therefore, demand targeted efforts.[7] For example, compared with women from the least deprived populations, investigations in England have shown that women living in more deprived areas may not only be more likely to be overweight or living with obesity and report the misuse of illicit drugs and smoking, but they may also be less likely to stop smoking during pregnancy, take folic acid supplements and book their first antenatal appointment within 10 weeks of pregnancy as recommended.[8] A comprehensive list of preconception health risk factors is provided in online supplemental appendix I.

Preconception care plays a crucial role in the screening, prevention and management of risk factors that may affect maternal and infant outcomes.[9] The provision of preconception care includes the development and implementation of strategies, policies, services, interventions, campaigns, guidelines, frameworks and initiatives. Because preconception care operates at individual, community and population levels, it can positively impact both those who actively plan to conceive and those who do not. This is of particular interest because of the high prevalence of unplanned pregnancies; in the UK, 45% of pregnancies and one-third of births are unplanned or accompanied by feelings of ambivalence.[2 10]

Recent years have witnessed an increased recognition of the importance of preconception health and care,[6] in both the academic field and policy (eg, Making the Case for Preconception Care[2]), suggesting that recent publicly accessible resources may have been produced but not yet reviewed in this capacity.

Shawe *et al* conducted an investigation of preconception policies, guidelines, recommendations and services available in six different countries, including the UK, to provide a baseline comparison of current materials.[9] They found variations between recommendations (eg, folic acid supplementation) and concluded that guidelines addressing the preconception phase are overall heterogeneous.[9] As these previous findings are related to searches conducted in 2013, a renewed investigation is now warranted covering the past decade.

A further study on preconception guidelines, recommendations and policy reports was conducted in 2021.[6] The review, however, intended to inform the reporting of population-level preconception health in England and identify preconception indicators, which then formed the basis for the development of a national surveillance system for preconception health in England.[11] Sixty-six indicators were identified across 12 domains, namely wider determinants of health, healthcare, environmental exposures, reproductive health and family planning, health behaviours, cervical screening, immunisation and infections, mental and physical health, medications and genetic risk.[6]

A systematic review was recently conducted to analyse freely accessible international clinical guidelines focusing on preconception care.[12] They identified eleven guidelines and assessed the quality of the guidelines themselves using the Appraisal of Guidelines for Research & Evaluation Instrument (AGREE II), the overall recommendations and the evidence recommendations are based on.[12] Findings suggested that high-quality guidelines are lacking and that guidelines should be improved by broadening the comprehensiveness of the content areas addressed, applying a more rigorous development process and enhancing the acceptability and feasibility of their application.[12]

No previous review was found that included both proposed locations of this scoping review: UK and Ireland.

## Objectives

From the available literature, it is clear that at present, it is important to collate the evidence in a coherent and accessible way that enables the identification of common themes, concepts and recommendations, priority areas not yet attended to and strengths and weaknesses of preconception care actions in place across the UK and Ireland. This scoping review aims to build on existing research and map the evidence in relation to preconception health and care strategies, policies, guidelines, frameworks and recommendations in the UK and Ireland, by summarising recurring themes and concepts underpinning the evidence, identifying gaps in knowledge and exploring future research priorities. This review will also clarify the type of evidence available and the target audience/s of included resources. Northern Ireland will be treated as a case study and, therefore, data on the services, interventions and initiatives provided in the region will be examined, presented and supplemented by consultations and audits with relevant stakeholders.

A systematic scoping approach will be used, as it is appropriate to map the principal themes of a broad research area of interest, describe the breadth of evidence available and synthesise heterogeneous evidence.[13] The present protocol will ensure transparency in the research process and overall reliability.[14]

Findings from this review will ultimately inform future research, practice and decision-making to optimise preconception health and care, and may help the development of a clear pathway for the promotion of evidence-based advice and support for individuals of childbearing age. In addition, findings may aid the planning of preconception care actions, potentially leveraging and maximising existing public health programmes.

## Research questions

This scoping review aims to answer the following questions:

a. What strategies, policies, guidelines, frameworks and recommendations have been developed that address preconception health and care for adults in the UK and Ireland between January 2011 and May 2022?
b. What are the main concepts and themes underpinning strategies, policies, guidelines, frameworks and recommendations that address preconception health and care for adults in the UK and Ireland?
c. How does the evidence from strategies, policies, guidelines, frameworks and recommendations that address preconception care for adults differ across the UK and Ireland?
d. What are, if any, the gaps in the knowledge provided in strategies, policies, services, guidelines, frameworks and recommendations that address preconception health and care for adults in the UK and Ireland, and what areas require further coverage and inquiry?
e. What are, if any, the services and interventions provided in Northern Ireland focused on improving preconception health and care in adults?

Answering these questions will provide insights into how the topic of preconception health and care is addressed in the grey literature, summarise the evidence and present an overview of how evidence may differ across the selected countries.

## METHODS AND ANALYSIS

The proposed scoping review will be conducted in accordance with the Joanna Briggs Institute (JBI) updated methodological guidance for scoping reviews,[14] Arksey and O'Malley's framework for conducting scoping studies[13] and reported in line with the Preferred Reporting Items for Systematic Reviews and Meta-analyses extension for Scoping Reviews (PRISMA-ScR).[15]

A preliminary search of CINAHL, Web of Science, JBI Evidence Synthesis and BMJ Open was conducted and no current or underway scoping reviews on the topic were identified.

### Eligibility criteria

The recent growing interest in preconception health and attention paid to the optimisation of health before pregnancy has led to an increase in preconception care actions by governments and public health organisations (eg, folic acid fortification), meaning that there is evidence to meet the inclusion criteria for this scoping review.

To develop the eligibility criteria, the Participants–Concept–Context framework outlined by Peters *et al* was used,[16] as described below. These criteria were developed in accordance with the study objectives and informed by the meaningful findings highlighted by Schoenaker *et al*[6] and the critical discourse presented in resources such as Making the Case for Preconception Care.[2]

### Participants

This review will consider resources discussing or reporting on preconception health and care for individuals of childbearing age, and no exclusions in relation to gender, ethnicity, culture, sexuality, health condition/s or disability status will be applied. Therefore, resources addressing individuals with chronic diseases, such as diabetes and epilepsy, will be considered for inclusion. Relevant resources adopting a life-course approach, thus addressing society as a whole, will be considered for inclusion as they also encompass individuals of childbearing age.

### Concept

The present work will summarise evidence underpinning strategies, policies, guidelines, frameworks and recommendations in the UK and Ireland regarding preconception health and care (see online supplemental appendix II Glossary of terms), including topics such as fitness for pregnancy, pregnancy planning and preparation for pregnancy. Furthermore, it will review services and interventions in Northern Ireland, as an individual case study. A case study approach allows an in-depth analysis of phenomena in a given context.[17]

Resources explicitly addressing only the interconception period (see online supplemental appendix II Glossary of terms) will not be considered eligible, also to avoid duplication of research due to a recent policy review covering this specific period.[18]

### Context

The review will only include resources relevant to the UK and Ireland. Whenever it is unclear whether a resource is relevant, the author/s or relevant organisation/s will be contacted.

### Types of sources

This scoping review will consider grey literature sources not found in the published literature. In this case, the grey literature included refers to strategies, policies, guidelines, frameworks and recommendations addressing preconception health and care in the UK and Ireland. Technical or research reports from government agencies, registered charities or scientific research groups, documents outlining interventions or initiatives from public bodies, articles and guidelines issued by government agencies or professional bodies will be considered for inclusion. In addition, leaflets and educational booklets will be considered for inclusion, as well as relevant e-learning resources due to their increasing popularity among healthcare professionals and ability to provide information that is easily understandable.

Journal articles, preprints (journal articles not yet peer-reviewed or published), working papers from research groups, visual or audio content such as television programmes or documentaries (or reviews of this content), academic letters or commentaries, calls for participants, presentations and doctoral dissertations will not be considered for inclusion in this scoping review.

Grey literature, including but not limited to policies, was deemed of interest as opposed to published literature

as it is directed at the public, patients, healthcare professionals and governments, who are the ones who may act directly as a response to recommendations in the documents to review.[6] Grey literature holds the potential to disseminate findings to wider audiences as it is largely accessible also to non-specialist audiences,[19 20] and can provide a current, balanced and comprehensive view of the evidence.[21]

Overall, to be considered for inclusion, the material will need to provide concrete, tangible advice and guidance, deliver recommendations or alternatively outline policy actions or strategic plans to improve preconception health and care for adults of childbearing age. Whenever a resource solely mentions preconception health or simply signposts other material, it will not be eligible for inclusion. Although this review aims to summarise grey literature, certain strategies, policies, guidelines, frameworks and recommendations identified during the searches may also be published in the literature.

Details of the inclusion and exclusion criteria applied are presented in online supplemental appendix III.

### Search strategy
Initial searches were undertaken on Google Advanced Search in March and April 2022 to develop the search strategy for database searches. A subject librarian was also consulted to build a transparent and robustly structured search strategy (see online supplemental appendix IV).

The search strategy was then applied in May 2022 using the databases Google Advanced Search, National Institute for Health and Care Excellence (NICE), OpenAire and ProQuest. Public health and government websites were also searched (see online supplemental appendix V). This process of locating sources was informed by Godin *et al*.[22] Other grey literature databases such as EThOS, DART, Grey Literature Report, OpenDoar were explored but excluded as not deemed relevant for the study. Boolean operators OR and AND were used. The search strategy, including all identified keywords and index terms, was adapted for each database and/or information source. The reference list of included sources of evidence will be screened for additional sources. Consultations with experts will be conducted in Spring 2023 to inform research and contextualise findings. Therefore, they will be conducted once preliminary findings from the scoping review are collated.

Only documents published in English will be included, as the inquiry is limited to primarily English-speaking countries. Resources published, reviewed or updated during or after January 2011 will be considered for inclusion, building on the time frame of a previous review across six European countries[9] and allowing for more than a decade of content to be considered. Material from UK-based and/or Ireland-based Royal Colleges and charitable organisations and associations with a location or contact address in one of the devolved nations will be deemed eligible for inclusion. Whenever a potentially relevant resource is not readily available or access is restricted, the relevant body or authors will be contacted.

### Case study
Consultations and audits with relevant stakeholders are favourable to validate findings and ensure breadth of coverage.[13] The searches specific to Northern Ireland will be supplemented by audits with stakeholders, including experts in the field of preconception and healthcare professionals (eg, maternity service providers and clinical leads, consultant midwives, pharmacists). Northern Ireland was deemed of interest as a case study because, although part of the UK, its devolved government has the ability to set its own policies, legislations and agenda. Moreover, preconception health is increasingly receiving importance in Northern Ireland, which has led to the development of a Strategy for maternity care,[23] increased funding for the development of perinatal mental health teams across Health and Social Care Trusts[24] and support for the introduction of the mandatory fortification of flour with folic acid, for example.[25]

Overall, these consultations aim to identify other potentially eligible resources, understand any under-researched issues concerning the effectiveness of services or strategies and contextualise findings. Stakeholders will be sent information about the project prior to a consultation and will be asked to comment on preliminary findings and on whether any relevant resources may be missing. Questions and topic areas discussed during audits will be informed by the findings of the searches conducted. Contributors will also receive communications about the findings of the study, once completed.[26]

### Selection of sources of evidence
Citations identified through searching will be collated and uploaded into Microsoft Excel and duplicates will be removed. Titles and summaries will then be screened by two independent reviewers for assessment against the inclusion criteria for the review. Following the methodology used by Godin *et al* for examining grey literature,[22] only the first 100 results on Google Advanced Search will be reviewed for potentially relevant titles. Potentially relevant sources will be retrieved in full, and the full texts will be assessed in detail against the inclusion criteria by two independent reviewers. At least 10% of resources will be double-coded. Any disagreements that arise between the reviewers at each stage of the selection process will be resolved through discussion, or with an additional reviewer/s, until consensus is reached. Reasons for the exclusion of full-text sources of evidence that do not meet the inclusion criteria will be recorded and reported in the scoping review. Overall, the results of the search and the study inclusion process will be reported in full in the final scoping review and presented in a PRISMA-ScR flow diagram.[15] The diagram will illustrate where citations were discarded during the screening process, and a rationale for the exclusion of resources during the full-text screening will also be provided.

## Data extraction

Data will be extracted by the same authors who performed the screening and full-text review to ensure consistency. Data will be extracted from eligible resources independently by two reviewers using a data extraction tool developed by the reviewers on Excel. The data extracted will include specific details about the publication title, publication format (eg, report, guideline, strategy), target audience (eg, healthcare professionals, policy-makers), overarching aim (eg, increase awareness among individuals of childbearing age, educate healthcare professionals), participants (eg, individuals of childbearing age, individuals of childbearing age with diabetes), geographical location, year of publication, review or updates, duration of the strategy, policy or intervention when applicable, and key notes relevant to the research questions. These extracted values were agreed on after pilot testing the draft extraction form between authors (initial example shown in online supplemental appendix VI). The revised data extraction instrument (online supplemental appendix VII) will be amended if necessary during the data extraction process. The modifications will be detailed in the full scoping review. Any disagreements between the reviewers will be resolved through discussion, or with an additional reviewer/s. When required, an attempt will be made to contact the authors of original resources for clarification if data are unclear, or to request missing or additional data. A follow-up email will be sent after 2 weeks from the initial contact if no response is received. Thereafter, the missing or unclear data will be documented in the review as such. A critical appraisal of individual sources of evidence will not be undertaken, as this is not a requirement for scoping reviews and is outside the scope of this review. A further reason not to undertake critical appraisal was the diversity of the grey literature considered for inclusion. However, the sources of information of the reviewed documents will be inspected, to provide insight on the credibility and accuracy of the advice provided.

## Data analysis and presentation

The characteristics of included resources will be described in detail (eg, participants, target audience), a distribution of studies by year will be produced and a map of the available evidence will be presented in tables. Comparisons will be made based on the included evidence across the UK and Ireland, and resources addressing preconception health and care services, interventions and initiatives in Northern Ireland will be presented as a case study. Included resources will be coded using NVivo, and at least 10% of the data will be double-coded. A narrative synthesis will highlight the emerging themes and concepts and emphasise how the findings relate to the research questions. The limitations and research gaps will be emphasised and discussed. Authors will adhere to PRISMA-ScR when reporting findings.[15]

## Patient and public involvement and engagement

The proposed review actively involves a patient and public involvement and engagement (PPIE) panel of adults aged 18–45 years living in Northern Ireland, named 'Healthy Reproductive Years'. The PPIE strategies aim to engage the public as partners and mobilise their existing knowledge and expertise, communicate using a lay language and support a collaborative approach in research by fostering respect and honesty.[26] The PPIE advisory panel was recruited via social media platforms (eg, Facebook), relevant organisations (eg, Sure Start), charities and leisure and community centres in Northern Ireland. Recruitment activities aimed to achieve diversity in terms of gender, sexuality, ethnicity, religion, disability, socioeconomic background, education level and health literacy. The PPIE panel was developed to contribute to general discussions and advise on research design, priority setting and dissemination plans which will include the publication of the final report and the coproduction of conference materials, a lay summary and an infographic presenting review findings. Representatives have been integrally involved in the development of this protocol and have provided direct feedback on all aspects of it. Collaboration with the PPIE panel will occur via online or in-person meetings and exchange of emails or postal letters.

The reporting of PPIE strategies in this research project will be guided by the Guidance for Reporting Involvement of Patients and the Public (GRIPP) 2 checklist.[27] All PPIE representatives will be reimbursed for their contributions.

## ETHICS AND DISSEMINATION

Ethical approval is not applicable because this scoping review will analyse publicly accessible resources only. The consultations and audits with key stakeholders conducted to validate findings relative to Northern Ireland do not require ethical approval as these are only intended as tools to inform research. Findings will be disseminated to service users and relevant stakeholders using diverse approaches, targeted to recipients. Outcomes following this review include a peer-reviewed publication reporting a detailed synthesis of findings and comparisons across countries, a lay summary, a clear and accessible infographic, and presentations directed at conference audiences, stakeholders and the public. The PPIE 'Healthy Reproductive Years' panel will be consulted and will inform the dissemination plan. In addition, a workshop will be held with PPIE representatives to discuss findings.

## DISCUSSION

Recent years have seen the rise in the publication of key documents addressing preconception health and care that have the opportunity to positively influence the current landscape by supporting positive health changes, including Making the Case for Preconception

Care,[2] Missed Periods,[28] A Strategy for Maternity Care in Northern Ireland[23] and Women's Health Strategy for England.[29] These documents emphasise the need to build the evidence base, raise awareness on women's health and preconception health and care, improve information-sharing and ultimately ensure that all adults of reproductive age are given the tools to optimise their preconception physical, social and psychological health. Recently, it has also been announced that folic acid will be added to non-wholemeal wheat flour in the UK as a way to reduce the incidents of neural tube defects.[25] It is also important to acknowledge that efforts have been made globally. Examples include a policy brief published by the WHO with the aim of improving preconception care[3] and the identification of a condensed set of preconception health indicators in the USA with the aim of better monitoring the health status of women of reproductive age.[1]

Currently, no comprehensive synthesis of the evidence relating to this topic is available in the proposed context of the UK and Ireland. Although this review has potential strengths, such as the transparency of the approach adopted, the addition of consultations with experts and the involvement of a PPIE advisory panel, there are also challenges and limitations, such as the broad scope of the review and the lack of a formal critical appraisal of the individual resources. Both strengths and limitations will be clearly explored and further highlighted in the full report.

**Acknowledgements** The authors would like to acknowledge the contribution of the patient and public involvement and engagement 'Healthy Reproductive Years' panel for their assistance.

**Contributors** EHC, LM, MCM and LK discussed the protocol and finalised the methodological process of the scoping review. EHC drafted the protocol, which was finalised with input from LM, MCM, LK, K-AE and DAJMS. DT and TS, members of the patient and public involvement and engagement 'Healthy Reproductive Years' panel, also helped shape the research and reviewed the protocol. All authors have seen and approved the final version of the protocol for publication.

**Funding** This work is supported by Queen's University Belfast as part of a PhD studentship for EHC. EHC is funded by the Department for the Economy NI. DAJMS is supported by the National Institute for Health and Social Care Research (NIHR) Southampton Biomedical Research Centre (SBRC-1215-20004).

**Disclaimer** The views expressed are those of the author(s) and not necessarily those of the NIHR or the Department of Health and Social Care.

**Competing interests** None declared.

**Patient and public involvement** Patients and/or the public were involved in the design, or conduct, or reporting, or dissemination plans of this research. Refer to the Methods section for further details.

**Patient consent for publication** Not applicable.

**Provenance and peer review** Not commissioned; externally peer reviewed.

**ORCID iDs**
Emma H Cassinelli http://orcid.org/0000-0001-8778-0801
Michelle C McKinley http://orcid.org/0000-0003-3386-1504
Lisa Kent http://orcid.org/0000-0002-8882-0526
Kelly-Ann Eastwood http://orcid.org/0000-0003-3689-0490
Danielle A J M Schoenaker http://orcid.org/0000-0002-7652-990X
Laura McGowan http://orcid.org/0000-0003-3253-3226

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
