## [Reviewer comments · BMJ Open]

ARTICLE DETAILS

TITLE (PROVISIONAL)	Preconception health and care policies, strategies and guidelines in the UK and Ireland: A scoping review protocol
AUTHORS	Cassinelli, Emma; McKinley, Michelle C.; Kent, Lisa; Eastwood, Kelly-Ann; Schoenaker, Danielle; Trew, David; Stoikidou, Theano; McGowan, Laura

VERSION 1 – REVIEW

REVIEWER	Oludoyinmola Ojifinni University of the Witwatersrand Johannesburg
REVIEW RETURNED	01-Nov-2022

GENERAL COMMENTS	The study is an important one and the outcomes will be useful to update preconception care guidelines in the UK which may be applicable to other parts of Europe and beyond. There are a few things that I believe the authors should address to improve the protocol. However, considering that the authors have identified previous reviews covering similar topics, the justification for the current review needs to be more substantial. For instance, what additional information is there now that wasn't included in the existing reviews? Have the guidelines described in those reviews been updated since the reviews were published? Are there any countries that had no documented policies, guidelines or strategies which now have them? With regards to the study design, this review seems more in keeping with a policy or textual review that with a scoping review especially as it is focused on grey literature. The rationale behind excluding journal articles needs to be clearer, otherwise the authors can consider a policy review as that is more suited to the methodology described. Guidance for this is available in the JBI manual of evidence synthesis. On the need for ethical clearance, the authors need to confirm that the case study of Ireland will not require this as there will be interviews of stakeholders as described in the methods. Concerning the strengths and limitations of the study, the list is largely generic and does not specify how the identified issues relate to the current study clearly.
--

REVIEWER	Winifred Chinyere Ukoha University of KwaZulu-Natal
REVIEW RETURNED	14-Nov-2022

GENERAL COMMENTS	Thank you for the opportunity to review this manuscript which addresses the Preconception health and care policies, strategies, and guidelines in the UK and Ireland. Please find attached below, some comments to assist the authors in strengthening the manuscript
---

REVIEWER	Sowmiya Moorthie University of Cambridge, Cambridge Public Health
REVIEW RETURNED	16-Jan-2023

GENERAL COMMENTS	The authors have presented a well written protocol for a scoping review on preconception health and care policies, strategies and guidelines in the UK and Ireland. This is an interesting study, addressing current practice and the evidence base. I had some minor comments. 1. The introduction covers some details about the risk factors, however, as this is a broad area, and while it is important to mention risk factors, given the focus of the review the level of detail seems incongruous. It may be better to shorten some of htis and expand on the later section (i.e. pg 2 line 44 onwards). This sets the practical context for conducting this review and also the relevant literature in this area. 2. A minor point - You could consider removing lines 24-29 on pg 5. Maybe a brief description of the Participant, context, framework and why this was chosen could be included. 3. pg 6, lines 12-18. Would be it be best to just state that resources explicitly addressing the interconception period will not be eligible due to a recent review? The authors state that they want to focus on the time before the first pregnancy, however, given the preceding definitions, and also for practical reasons, to what extent can the focus be on time before the first pregnancy? pg 6 - Types of sources: The first few sentences don't quite make sense. The emphasis on the grey literature is because of the objectives of the review to look at current practice. It may be best to start with describing the types of grey literature that the review aimed at identifying. pg 7 - A minor point. Line 12/13. It says "...improve preconception health and care for adults." Previously it was those of child bearing age. Inclusion of e-learning resources - Could you please explain the rationale for including these groups (lines 52-53). Will they be analysed separately?
---

VERSION 1 – AUTHOR RESPONSE

Reviewer: 1

Dr. Oludoyinmola Ojifinni, University of the Witwatersrand Johannesburg Comments to the Author: The study is an important one and the outcomes will be useful to update preconception care guidelines in the UK which may be applicable to other parts of Europe and beyond. There are a few things that I believe the authors should address to improve the protocol.

However, considering that the authors have identified previous reviews covering similar topics, the justification for the current review needs to be more substantial. For instance, what additional

information is there now that wasn't included in the existing reviews? Have the guidelines described in those reviews been updated since the reviews were published? Are there any countries that had no documented policies, guidelines or strategies which now have them? Greater information has been added about previous reviews in the Introduction. Moreover, we have now clearly stated how this piece of work aims to build upon pre-existing knowledge by setting novel aims.

“Recent years have witnessed an increased recognition of the importance of preconception health and care[6], in both the academic and policy-related fields (e.g., Making the Case for Preconception Care[2]), suggesting that recent publicly accessible resources may have been produced but not yet reviewed in this capacity.

Shawe and colleagues conducted an investigation of preconception policies, guidelines, recommendations and services available in six different countries, including the UK, to provide a baseline comparison of current materials [9]. They found variations between recommendations (e.g., folic acid) and concluded that guidelines addressing the preconception phase are heterogeneous[9]. As these previous findings are related to searches conducted in 2013, a renewed investigation is now warranted covering the past decade.

A further study on preconception guidelines, recommendations and policy reports was conducted in 2021[6]. The review, however, intended to inform the reporting of population- level preconception health in England and identify preconception indicators, which then formed the basis for the development of a national surveillance system for preconception health in England[11]. Sixty-six indicators were identified across 12 domains, namely wider determinants of health, health care, environmental exposures, reproductive health and family planning, health behaviours, cervical screening, immunisation and infections, mental and physical health, medications and genetic risk[6]. A systematic review was recently conducted to analyse freely accessible international clinical guidelines focusing on preconception care[12]. They identified eleven guidelines and assessed the quality of the guidelines themselves using the AGREE II tool, the overall recommendations and the evidence recommendations are based on[12]. Findings suggested that high-quality guidelines are lacking and that guidelines should be improved by

broadening the comprehensiveness of the content areas addressed, applying a more rigorous development process and enhancing the acceptability and feasibility of their application[12].

No previous review was found that included both the proposed locations of this scoping review: UK and Ireland. “

“This scoping review aims to build upon existing research and map the evidence in relation to preconception health and care strategies, policies, guidelines, frameworks and recommendations in the UK and Ireland [...]”.

With regards to the study design, this review seems more in keeping with a policy or textual review than with a scoping review especially as it is focused on grey literature. The rationale behind excluding journal articles needs to be clearer, otherwise the authors can consider a policy review as that is more suited to the methodology described. Guidance for this is available in the JBI manual of evidence synthesis. Further information was added in regard to our selection of grey literature, as opposed to journal articles.

“Grey literature, including but not limited to policies, was deemed of interest as opposed to published literature as it is directed at the public, patients, healthcare professionals and governments, who are the ones who may act directly as a response to recommendations in the documents to review[6]. Grey literature holds the potential to disseminate findings to wider audiences as it is largely accessible also to non-specialist audiences[19, 20], and can provide a current, balanced and comprehensive view of the evidence[21].”

On the need for ethical clearance, the authors need to confirm that the case study of Ireland will not require this as there will be interviews of stakeholders as described in the methods. In the Ethics and Dissemination section a sentence added to clarify this.

“The consultations and audits with key stakeholders conducted to validate findings relative to Northern Ireland do not require ethical approval as these are only intended as tools to inform research.” (please see the Health and Social Care Research & Development Division Northern Ireland: <https://research.hscni.net/approval-research-hsc>).

Concerning the strengths and limitations of the study, the list is largely generic and does not specify how the identified issues relate to the current study clearly. Further strengths and limitations of the study have now been added.

“Although this review has potential strengths, such as the transparency of the approach adopted, the addition of consultations with experts and the involvement of a PPIE advisory panel, there are also challenges and limitations, such as the broad scope of the review and the lack of a formal critical appraisal of the individual resources. Both strengths and limitations will be clearly explored and further highlighted in the full report.”

Reviewer: 2

Winifred Chinyere Ukoha, University of KwaZulu-Natal Comments to the Author:

Thank you for the opportunity to review this manuscript which addresses the Preconception health and care policies, strategies, and guidelines in the UK and Ireland. Please find below some comments to assist you in strengthening your manuscript:

1. Why does the second objective lend itself to a scoping review approach? Many thanks for your comment. We believe that scoping reviews are appropriate for assessing the “breadth rather than depth” of evidence and for synthesising and interpreting data according to key issues and themes (see: doi:10.1080/1364557032000119616). Additionally, the data extraction tool was tailored to this scoping review to aid the process (e.g., to clarify the type of evidence available).

2. What is already known about PCHC intervention and initiatives in Northern Ireland? A sentence was added in the “Case Study” section. “Moreover, preconception health is increasingly receiving importance in Northern Ireland, which has led to the development of a Strategy for maternity care[23], increased funding for the development of perinatal mental health teams across Health and Social Care Trusts[24] and support for the introduction of the mandatory fortification of flour with folic acid, for example[25].”

3. Page 5 line 31, “was used”? Thank you for your observation. We believe that a past tense is appropriate here, to refer to something already performed. In this case, the Participants- Concept-Context framework was used to streamline the eligibility criteria, as described in the following paragraphs.

4. Page 6, lines 13 – 18 amend this and just tell us why you have to exclude interconception care here. There should be a section for the definition of terms. This section has been amended, also respecting another reviewer’s comments. Appendix II was added to include a table titled “Glossary of terms”. The definitions in-text were amended as suggested.

“Resources explicitly addressing the interconception period (see Appendix II Glossary of terms) will not be considered eligible, also to avoid duplication of research due to a recent policy review covering this specific period[18].”

5. Page 8 “selection of the source of evidence”. Indicate how you will locate the “grey literature” as most of them are not published. The fact that evidence is not published does not influence the selection of evidence, as we aim to analyse publicly accessible resources only. We recognise that a limitation of the study is that we will only include resources available online; however, in the context of Northern Ireland, the additional case study element will help overcome this by identifying any other resources that might have been missed. We appreciate your observation and we have tried to make this clearer with the following sentences:

“Searches were carried out in May 2022 using the databases Google Advanced Search, National Institute for Health & Clinical Excellence (NICE), OpenAire and ProQuest. Public health and

government websites were also searched (see Appendix V). This process of locating sources informed by Godin et al.[22].”

“[...] because this scoping review will analyse publicly accessible resources only”. 6. What is the time frame for the SCR? Greater detail on the time frame has now been included.

“Searches were carried out in May 2022 [...]”.

“Consultations with experts will be conducted in March and April 2023, as a means to inform research and contextualise findings. Therefore, they will be conducted once preliminary findings from the scoping exercise are collated.”

7. State the rationale for the eligibility criteria. The relevant sentence was amended, and additional information was added to clarify the rationale.

“The recent growing interest in preconception health and attention paid to the optimisation of health before pregnancy has led to an increase in preconception care actions by governments and public health organisations (e.g., mandatory folic acid fortification), meaning that there is evidence to meet the inclusion criteria for this scoping review.

To develop the eligibility criteria, the Participants-Concept-Context framework outlined by Peters and colleagues was used[16], as described below. These criteria were developed in accordance with the study objectives and informed by the meaningful findings highlighted by Schoenaker and colleagues[6] and the critical discourse presented in resources such as Making the Case for Preconception Care [2].”

8. Present the result of the pilot searches. We appreciate the comment. Pilot searches were only conducted to identify search terms and values to include in the data extraction tool. An example has now been added in Appendix VI.

9. Give the motivation for omitting critical appraisal. The sentence was changed for clarity purposes. “A critical appraisal of individual sources of evidence will not be undertaken, as this is not a requirement for scoping reviews and is outside the scope of this review. A further reason not to undertake critical appraisal was the diversity of the grey literature considered for inclusion. However, the sources of information of the reviewed documents will be inspected, to provide insight on the credibility and accuracy of the advice provided.” Overall, we believe that critical appraisal may be more relevant for studies aiming to look in depth at the literature (e.g., systematic reviews), instead of the breadth

(see: https://www.tandfonline.com/doi/pdf/10.1080/1364557032000119616?casa_token=jbL09UGPWBkAAAAA:DdLbsjuKw3pBKhek1hTbMQ-LmP7WsUqhedYmfb8MtuIT24It9f1UOVVG-aRJKpNlzmtqj7Jx4JaaHw). We appreciate, however, that once the evidence is mapped in this scoping review, a subsequent useful step would be to evaluate the evidence.

10. Where is the PRISMA chart? The PRISMA chart will be included in the main manuscript with results, once all the relevant resources have been identified and assessed against the eligibility criteria. The PRISMA chart will also show any additional resources identified through audits with experts, which have not yet been conducted (as of 23rd February 2023).

Reviewer: 3

Dr. Sowmiya Moorthie, University of Cambridge, PHG Foundation Comments to the Author:

The authors have presented a well written protocol for a scoping review on preconception health and care policies, strategies and guidelines in the UK and Ireland. This is an interesting study, addressing current practice and the evidence base. I had some minor comments.

1. The introduction covers some details about the risk factors, however, as this is a broad area, and while it is important to mention risk factors, given the focus of the review the level of detail seems incongruous. It may be better to shorten some of this and expand on the later section (i.e. pg 2 line 44 onwards). This sets the practical context for conducting this review and also the relevant literature in this area. Many thanks for the observation. The paragraph describing risk factors has now been shortened, and greater detail was added to expand the subsequent section, as suggested.

“Shawe and colleagues conducted an investigation of preconception policies, guidelines, recommendations and services available in six different countries, including the UK, to provide a baseline comparison of current materials [9]. They found variations between recommendations (e.g., folic acid) and concluded that guidelines addressing the preconception phase are heterogeneous[9]. As these previous findings are related to searches conducted in 2013, a renewed investigation is now warranted covering the past decade.

A further study on preconception guidelines, recommendations and policy reports was conducted in 2021[6]. The review, however, intended to inform the reporting of population- level preconception health in England and identify preconception indicators, which then formed the basis for the development of a national surveillance system for preconception health in England[11]. Sixty-six indicators were identified across 12 domains, namely wider determinants of health, health care, environmental exposures, reproductive health and family planning, health behaviours, cervical screening, immunisation and infections, mental and physical health, medications and genetic risk[6]. A systematic review was recently conducted to analyse freely accessible international clinical guidelines focusing on preconception care[12]. They identified eleven guidelines and assessed the quality of the guidelines themselves using the AGREE II tool, the overall recommendations and the evidence recommendations are based on[12]. Findings suggested that high-quality guidelines are lacking and that guidelines should be improved by broadening the comprehensiveness of the content areas addressed, applying a more

rigorous development process and enhancing the acceptability and feasibility of their application[12].”

2. A minor point - You could consider removing lines 24-29 on pg 5. Maybe a brief description of the Participant, context, framework and why this was chosen could be included. The description of the Participants-Concept-Context framework has been shortened, also in line with feedback from other reviewers. The framework was chosen as advised by Peters et al. in Chapter 11: Scoping Reviews of the JBI Manual for evidence synthesis, as referenced.

“To develop the eligibility criteria, the Participants-Concept-Context framework outlined by Peters and colleagues was used[16], as described below.”

3. pg 6, lines 12-18. Would it be best to just state that resources explicitly addressing the interconception period will not be eligible due to a recent review? The authors state that they want to focus on the time before the first pregnancy, however, given the preceding definitions, and also for practical reasons, to what extent can the focus be on time before the first pregnancy? We aim to exclude those resources that explicitly only address the interconception period. We appreciate that some resources may address both the preconception and interconception period, and this will be discussed further in the main manuscript, based on results.

The wording in the protocol has been changed to reflect the kind suggestion. “Resources explicitly addressing the interconception period (see Appendix II Glossary of terms) will not be considered eligible, also to avoid duplication of research due to a recent policy review covering this specific period[18].”

pg 6 - Types of sources: The first few sentences don't quite make sense. The emphasis on the grey literature is because of the objectives of the review to look at current practice. It may be best to start with describing the types of grey literature that the review aimed at identifying. The description of the types of grey literature considered eligible has now been moved at the beginning of the “Types of Sources” section, as suggested.

A brief definition of grey literature and the rationale for inclusion has been retained, in response to another reviewer's comments and as a way to balance comments from all reviewers.

pg 7 - A minor point. Line 12/13. It says "...improve preconception health and care for adults."

Previously it was those of child bearing age. The sentence was amended to avoid confusion. "... to improve preconception health and care for adults of childbearing age." Inclusion of e-learning resources - Could you please explain the rationale for including these groups (lines 52-53). Will they be analysed separately? Greater detail was included regarding the rationale for inclusion of e-learnings. This sentence has been changed to: "... leaflets and educational booklets will be

considered for inclusion, as well as relevant e-learning resources due to their increasing popularity among healthcare professionals and ability to provide information that is easily understandable.”

While findings on the e-learning will not be analysed separately, their inclusion will be highlighted in the main manuscript.

VERSION 2 – REVIEW

REVIEWER	Oludoyinmola Ojifinni University of the Witwatersrand Johannesburg
REVIEW RETURNED	22-Mar-2023
GENERAL COMMENTS	A minor point - is the search reported to have been conducted (in the abstract and methods) the main search for the study or a pilot to identify relevant information with regards to the search strategy?

VERSION 2 – AUTHOR RESPONSE

Reviewer: 1

Dr. Oludoyinmola Ojifinni, University of the Witwatersrand Johannesburg

Comments to the Author:

A minor point - is the search reported to have been conducted (in the abstract and methods) the main search for the study or a pilot to identify relevant information with regards to the search strategy? The dates of the search which are stated in the abstract and methods section (i.e., May 2022) refer to the main search for the study. Previous searches were conducted to inform the search strategy. In the search strategy section of the protocol, sentences have been amended to better clarify this: “Initial searches were undertaken on Google Advanced Search in March and April 2022 to develop the search strategy for database searches. [...] The search strategy was then applied in May 2022 using the databases Google Advanced Search, National Institute for Health & Clinical Excellence (NICE), OpenAire and ProQuest.”